# Optimization of *Aedes albopictus* (Diptera: Culicidae) Mass Rearing through Cost-Effective Larval Feeding

**DOI:** 10.3390/insects13060504

**Published:** 2022-05-26

**Authors:** Mihaela Kavran, Arianna Puggioli, Sara Šiljegović, Dušan Čanadžić, Nikola Laćarac, Mina Rakita, Aleksandra Ignjatović Ćupina, Fabrizio Balestrino, Dušan Petrić, Romeo Bellini

**Affiliations:** 1Faculty of Agriculture, University of Novi Sad, Trg Dositeja Obradovića 8, 21000 Novi Sad, Serbia; mihaela.kavran@polj.edu.rs (M.K.); sarasiljegovic@yahoo.com (S.Š.); canedusan9@gmail.com (D.Č.); nlacarac@gmail.com (N.L.); minarakita.mr@gmail.com (M.R.); dusanp@polj.uns.ac.rs (D.P.); 2Sanitary Entomology & Zoology Department, Centro Agricoltura Ambiente “G. Nicoli”, IAEA Collaborating Center, Via Sant’Agata 835, 40014 Crevalcore, Italy; apuggioli@caa.it (A.P.); fbalestrino@caa.it (F.B.); rbellini@caa.it (R.B.)

**Keywords:** *Aedes albopictus*, SIT, larval diet, mass rearing, IAEA-BY, BCWPRL, MIX-14

## Abstract

**Simple Summary:**

The Asian tiger mosquito (*Aedes albopictus*) is an important invasive species of medical concern, which could be successfully suppressed by including the sterile insect technique (SIT) in integrated mosquito management. This technique is based on the mass rearing of males, and their sterilization and release into the habitats to compete with wild males in the mating process. Our research compared the effectiveness of three larval diet recipes (IAEA-BY, BCWPRL, and MIX-14) in the rearing of *Ae. albopictus* males in order to evaluate the available economical feeding alternatives. The separation of male pupae was done by the sieving method, and reared adult males were tested for flight capacity and longevity. The application of BCWPRL resulted in a higher portion of sieved male pupae than females, but the obtained number of both pupae and adult males was lower and the development was slower than the other two diets. The adult mean survival time was the highest in males fed with MIX-14 and the lowest in males fed with IAEA-BY. Males fed by IAEA-BY also demonstrated higher initial mortality in the adult stage. The diets BCWPRL and MIX-14 are cheaper than IAEA-BY (2.28 and 5.30 times, respectively). The diet MIX-14 represents a candidate for replacing the effective but still expensive IAEA-BY diet.

**Abstract:**

*Aedes* (*Stegomyia*) *albopictus* (Skuse, 1895) is an invasive important medical and veterinary pest species. The sterile insect technique (SIT) involves the mass rearing of males, and their sterilization and release into the habitat to compete with wild males. Our research objective was to compare the effectiveness of three larval diet recipes (IAEA-BY, BCWPRL, and MIX-14) in the laboratory rearing of *Ae. albopictus* males to evaluate the available economical feeding alternatives. The separation of sexes was done in the pupal stage by sieving. Reared males were tested for flight capacity and longevity. The application of the BCWPRL diet resulted in a higher portion of sieved male pupae than females, but the development of males was the slowest, and the number of obtained males (pupae and adults) was lower compared to the other two diets. The adult mean survival time was the highest in males fed with MIX-14 and the lowest in males fed with IAEA-BY. Males fed by IAEA-BY also demonstrated higher initial mortality in the adult stage. The diets BCWPRL and MIX-14 are economically more convenient than IAEA-BY (2.28 and 5.30 times cheaper, respectively). The cheapest diet, MIX-14, might represent a candidate for replacing the effective but still expensive IAEA-BY larval diet, providing lower costs of sterile male production.

## 1. Introduction

*Aedes* (*Stegomyia*) *albopictus* (Skuse, 1895) (Diptera: Culicidae)*,* the Asian tiger mosquito, has been rapidly spreading over the world in the past three decades. Originating from South East Asia and the islands of Western Pacific and Indian Ocean, it invaded Africa, the Americas, Australia, and Europe, including almost all countries therein [1,2,3,4]. With high adaptability, high reproductive potential, and vector competence for at least 20 arboviruses, it became the primary molestant of a public health concern at the areas where it was established. This species is the main vector of the Chikungunya virus, an important vector of the Dengue virus, which is less important as a vector of Zika virus, and it also demonstrated the potential to transmit the Usutu virus [5,6,7,8,9,10,11,12,13,14,15]. 

Efforts to suppress this species have been mainly relying on conventional control measures. Due to the complex biology and ecology of *Ae. albopictus,* treatments against this species have not provided the reduction in its populations to satisfyingly low levels so far. As a result of the persistent use of insecticides, this species has developed resistance to a wide range of pyrethroids [16,17,18,19,20,21,22,23]. As conventional mosquito control has often failed to suppress *Ae. albopictus*, it is necessary to introduce and maintain alternative methods of control. The sterile insect technique (SIT) has a desirable potential to suppress the *Ae. albopictus* population, which was demonstrated throughout several pilot studies [24]. The sterile insect technique is an environmentally friendly method, based on mass production, sterilization, and the release of sterile *Ae. albopictus* males to reduce the fertility of a field population [25]. Hence, for SIT control, it is necessary to rear a large number of target species individuals, sterilize them with gamma irradiation, and release them into the control area to mate with the wild population [25,26]. A high number of sterile males is required to successfully compete with wild males. The more sterile males of a good condition are released, the higher the effectiveness of SIT will be. One of the requirements of mass production is to keep the quality of sterile males consistently high. For that reason, in mass production larvae should be reared under optimal climate-controlled conditions and fed an adequate larval diet. Larvae need proteins/amino acids, fatty acids, nucleic acids, sterols, and vitamins to go through regular larval development [27]. Previous studies confirmed that the quality of a larval diet is directly proportional to the quality of reared adult individuals [28,29,30,31]. 

In the last few years, several studies have been conducted to evaluate, introduce, improve, and optimize the SIT against *Aedes* invasive mosquitoes [32,33,34,35]. Not only the cost of the larval diet but also the wide availability on the market and the standardization of the diet components are crucial to improve mass-rearing procedure. Therefore, it is essential to provide affordable and cheap larval diet mixtures that are still of a high nutritional quality to enable the sustainable production of healthy competitive males. The most expensive component in the standard IAEA-BY mosquito larval diet developed by the international atomic energy agency (IAEA) [36,37] is the dehydrated bovine liver powder. Several studies focused on the reduction or replacement of the most expensive components were carried out [36,38,39,40,41,42,43]. Although very efficient, diets containing bovine liver powder were too expensive for the continuous process of mass rearing. Bimbilé Somda et al. [39] and Mamai et al. [41] tested insect-based diets, which consisted of *Hermetia illucens* (Linnaeus 1758) [39,41], *Tenebrio molitor* Linnaeus 1758, and *Musca domestica* Linnaeus 1758 [39] powders, along with other regularly used components of the reference IAEA larval diet, and compared the effects of the different combinations of ingredients to the mass rearing of *Ae. albopictus* and *Ae. aegypti* (Linnaeus 1762). Their findings highlighted the fact that edible insects in some combinations with other components provide a satisfying level of mosquito (larval) production of the two experimental species and also could contribute to food security and environmental protection. On the other hand, diets based on insects are still not easily available on the market in many European countries and thus could not be considered as regular mosquito diets in the mass rearing process. 

Our study focuses on the testing of easily available and cheap diets that would be comparable to a IAEA-BY standard diet. In particular, this work focuses on comparing the cost and impact of larval diets with cheap (BCWPRL and MIX-14) and costly (IAEA-BY) ingredients on: (i) the pupation onset and dynamics (the pupation of males in a mass-rearing process should be synchronized), (ii) sex separation success in the sieving procedure (the size difference between male and female pupae should be expressed, enabling the males to pass through the sieve but not the females), (iii) the ratio of males to total pupae produced (the domination of male pupae should be expressed within 24 h of the pupation onset), (iv) male flight capacity, and (v) male longevity (after being released in the field, sterile males should compete with wild, fertile males). The results of the present study are to be confirmed on a large scale in a mass rearing facility, with the final goal to make sterile males’ production cheaper and area-wide SIT application more affordable. 

## 2. Materials and Methods

### 2.1. Test Mosquitoes

The *Aedes albopictus* mosquito colony, originating from the Podgorica region, Montenegro (MNE strain), has been maintained for 6 generations in the mass rearing facility of Centro Agricoltura Ambiente “Giorgio Nicoli” (CAA) in Crevalcore, Italy. Mosquitoes were reared in a climate room under controlled conditions (air temperature 28 ± 2 °C, relative humidity 80 ± 5%, and a photoperiod 14:10 h L:D), according the protocol described by Balestrino et al. [37,44]. Produced eggs were shipped by express courier to the laboratory of medical and veterinary entomology, University of Novi Sad, Serbia, in order to perform the present study. 

### 2.2. Larval Diets Composition

Three types of larval diets were tested: BCWPRL, MIX-14, and IAEA-BY, which consisted of the following ingredients: BCWPRL [45]: bean (Borlotti beans) 16.7% (grocery shop, Italy), corn 16.7% (grocery shop, Italy), wheat 16.7% (grocery shop, Italy), chickpea 16.7% (grocery shop, Italy), rice 16.7% (grocery shop, Italy), and bovine liver powder 16.7% (MP Biomedicals, Solon, OH, USA);MIX-14 [46]: tuna meal 70% (T. C. Union Agrotech, Bangkok, Thailand), brewer’s yeast 15% (MP Biomedicals, Solon, OH, USA), chickpea 15% (grocery shop, Italy), vitamin mix 0.46% *w*:*v*, and 4.6 g per liter solution (Vanderzant Vitamin Mix, Bio-Serv, Frenchtown, NJ, USA);IAEA-BY: a standard diet developed at the insect pest control laboratory (IPCL) of international atomic energy agency (IAEA) in Seibersdorf, Austria and CAA [37,38,42]: tuna meal 50% (T. C. Union Agrotech, Bangkok, Thailand), bovine liver powder 36% (MP Biomedicals, Solon, OH), brewer’s yeast 14%, (MP Biomedicals, Solon, OH, USA), and vitamin mix 0.2 % *w*:*v* 2 g per liter solution (Vanderzant Vitamin Mix, Bio-Serv, Frenchtown, NJ, USA).

For each diet, the dry powderized ingredients were mixed and then dissolved in water at a ratio of 50 g diet powder per liter of distilled water. Diet solutions were stored at 4 °C. 

### 2.3. Experimental Design

Three replicates were performed to test the effects of each of the three diets. For each experimental unit (replicate), an average of 4000 eggs were used (in total 36,000 eggs). The number of eggs was estimated by scanning the eggs laid on the filter paper and using an open-source image processing and analysis program (ImageJ, United States National Institute of Health) [43]. 

Egg hatching was performed in 1 l volume jars. In the afternoon hours, 700 mL of deionized water was poured into each of the nine jars, and papers with eggs were placed in the water (4000 of eggs/jar). In order to stimulate the hatching, 0.25 g of bacto nutrient broth (OXOID Ltd., Basingstoke, UK) and 0.05 g of yeast (Sigma-Aldrich Inc., St. Louis, MO, USA) were dissolved in 2 mL of water and added to each jar [43,44]. Jars were closed and placed for incubation at 31 °C during subsequent 15 h. 

The next day, nine trays (dimensions: 40 cm × 30 cm × 10 cm) were prepared and filled with 1.3 l of deionized water. The content of the jars (water with neonate larvae) was transferred into the trays (one jar/one tray) to achieve the total liquid volume of 2 L/tray and a larval density of approximately 2 larvae/mL of water. The diet amounts required per each tray and day were calculated based on the number of larvae, as described by Bellini et al. and Balestrino et al. [43,44]. On the first day, 16 mL of diet solution/tray was added; on the 2nd day, 32 mL; on the 3rd day, 48 mL; and on the 4^th^ day (last day of feeding), 64 mL. The water temperature was recorded in each experimental unit (tray) three times a day (at 9 AM, 12 h, and 15 PM) until the 6th post-hatching day. During the rearing procedure, the water temperature was not significantly different between the trays (see Appendix A). 

The exact time of pupation onset (the appearance of the first pupa) in each tray was recorded. The inspection was carried out from 9 h to 20 h. All pupae that appeared afterwards were considered as pupated the next day. The time elapsed from the immersion of eggs into the water to the appearance of the first pupa was taken in account to calculate the time to the earliest pupation.

Sex separation was carried out in the pupal stage by the sieving method described by Bellini et al. [43]. The sieving method is based on/exploits two natural phenomena: the protandry (male pupae develop faster than female pupae) and the size dimorphism (male pupae are generally smaller than female pupae). 

Sex separation was performed by using a sieve of 1400-µm mesh size (Retsch Test Sieve with steel mesh). Sieving was conducted 24 h after the first pupa appeared in any of the nine trays across all treatments. During the first 24 h after the pupation onset, the majority of formed pupae are expected to be males [43]. 

Juveniles (larvae and pupae) from each experimental unit (9 rearing trays) were collected from the tray by pouring the content (water with juveniles) through a dense net (the equipment and the application of the sieving method are presented in Appendix A). The water from the rearing tray was kept and returned to the tray initially used for rearing. The juveniles were immediately transferred from the net into the bucket for sieving, which was previously filled with tepid water (34 °C). After gentle stirring to move juveniles from the walls of the bucket, the sieve was inserted into the bucket, with the bottom (mesh) down, submerged 2 cm below the water surface. Juveniles were then kept in the bucket for the next three minutes to allow small pupae, assumed to be males, to the pass through the sieve in an attempt to reach the water surface for respiration. After 3 min, the sieve was taken out with the pupae that passed through the mesh. Those pupae (classified as “passed pupae”) were washed from the sieve into the new tray previously filled with water (enough to cover the sieve), then pipetted to glass beakers with 150 mL of water, to be placed afterwards in cages for adult rearing (dimensions: 30 × 30 × 30 cm; BugDorm 1; Mega View, Taichung, Taiwan).

Pupae that remained in the bucket (those that had not passed through the sieve, classified as “not-passed pupae”) were transferred to another new tray, pipetted into the glass beaker with 150 mL of water, and put in a separate cage to obtain the adults. 

Larvae were returned back to the initially used rearing tray with water containing dissolved larval diet (water kept after the netting) and left there for the next 2 days to develop to the pupal stage. Those larvae were not additionally fed and were checked for subsequent pupation in the next 2 days. 

The whole procedure of sieving was repeated with all the rearing trays. The exact numbers of pupae that passed and did not pass the sieve were recorded per each diet and replicate. 

In order to obtain adult mosquitoes for further different testing purposes, pupae that successfully passed through the sieve (category “passed pupae”) were divided in subgroups and placed separately in three cages. One cage with 70 pupae was aimed to obtain a sufficient number of adult males for flight capacity testing, the second cage with 120 pupae was aimed for adult male longevity evaluation, and the third cage the rest of “passed pupae” were placed and kept until the adult emergence. Adults that ecloded in all of the three cages were taken into account for the sex determination of the cumulative group “passed pupae”. 

In total, 27 cages were designated for rearing adults from pupae that passed through the sieve (3 variants of larval diet, 3 replicates, and 3 subgroups) and 9 cages for the rearing of adults from pupae that did not pass the sieve (3 variants of larval diet, 3 replicates).

The next day, newly developed pupae (48 h after the pupation onset) were pipetted directly from rearing trays in glass beakers (sieving was not carried out), counted, and placed in separate cages to emerge (9 cages in total). 

The same procedure was repeated the following day to harvest the newly developed pupae 72 h after the first pupation. At this point, individuals that still remained in the larval stage were excluded from further rearing. Emerged adults that developed from pupae collected at 48 h and 72 h after pupation onset were targeted for sex determination, exclusively. 

After eclosion, adults reared from both “passed” and “not-passed” categories of pupae (collected 24 h after pupation onset) were visually observed for sex determination. The number of male and female specimens was recorded in order to calculate the portion of males in both the “passed” and “not-passed” categories, as well as for the calculation of the total number and portion of males developed from pupae collected 24 h after pupation onset. Similarly, the portion of male pupae collected at 48 h and 72 h after pupation onset was calculated based on the number of emerged adults.

Adult males developed from pupae that successfully passed the sieving procedure (category “passed”) were used for the further testing of the two adult male performance parameters: longevity and flight capacity. During the experiment with adults, the average daily temperature was 23.79 ± 1.80 °C, RH 77.46 ± 10.79% and the photoperiod 14:10 h L:D. Males were provided with 10% sucrose solution from the moment of their eclosion. An adapted sugar feeder was placed inside each cage before the emergence of adults, providing the adults with ad libitum access to the sugar solution.

Flight capacity test: a flight testing device (FTD, Figure 1a), a recently designed quality control device developed in the insect pest control laboratory (IPLC) of IAEA [33], was used to assess male quality. Testing started after all males in all cages had emerged and when they were 2–4 days old. From each cage (replicate), 50 males were tested; they were introduced initially at the base of the device cylinder, which consisted of a series of tubes (Figure 1b). On the top of the device covered by the net, a BG lure was placed to encourage males to fly to the top, against the air current produced by the fan (Figure 1c). Males that passed through cylinder within a period of 2 h were considered as those that passed the test. The percentage of passed males out of the number of total males initially introduced in the cylinder was calculated. Males that did not pass the test remained inside the cylinder. Males produced from one replicate of each of the three diets were tested simultaneously (three test devices, Figure 1). The flight test was performed in three series to test all the replicates of the three diets.

The longevity test involved the following: three replicates of 120 male pupae per cage for each applied larval diet were placed in nine separate cages, as previously explained. Shortly after eclosion, adults were checked for female presence. All detected females in cages were removed and discarded. Cages were supplied with 10% sucrose solution in adapted sugar feeders, providing the males ad libitum access to the liquid feed. The bottoms of the cages were manually cleaned daily for sugar solution leakage, by using a clean tissue. The test was carried out until the last male died. A number of dead individuals were recorded on a daily basis, in the period from 10 to 11 AM. Dead adult males were removed from the cages and discarded. 

The cost of alternative and standard diets: to calculate the cost of three tested diets, the prices of ingredients given in the previous studies were taken [39,45,46].

### 2.4. Statistical Analysis

Data were analyzed by Statistica v.14.0.0.15 (1984–2020 TIBCO Software Inc., Palo Alto, CA, USA). Pupation onset (i.e., time to earliest pupation) was analyzed using the general linear model (GLM), comparing the means by the Tukey’s HSD test with significance level α *=* 0.05. We measured the time for each experimental unit, calculating the number of hours from the transfer of newly hatched larvae (L_1_) from jars into the rearing trays (the moment when they had the first contact with the diets) until the first pupa was formed. 

The efficacy of the three compared diets, reflected through the % of passed and not-passed pupae 24 h after the first pupation, as well as the % of pupae formed 48 h and 72 h after pupation onset, the portion of the male pupae obtained daily (24, 48, and 72 h after the first pupation), and flight capacity (the % of males that “escaped” from the testing cylinder). Obtained results were analyzed by the GLM after the arcsine sqrt transformation of the data expressed in percentages. 

The longevity of adult male mosquitoes reared on different diets mixtures was analyzed using Kaplan–Meier survival curves, and the mean survival of three different groups were compared using a Log-rank test in order to estimate the impact of the larval diet to the male adult life span.

## 3. Results

### 3.1. Time to Earliest Pupation

The time required to reach the pupal stage varied from a minimum of 95.5 h for IAEA-BY to a maximum of 102.5 h for BCWPRL. Although the IAEA-BY diet provided the shortest average pupation time (97.17 ± 2.08 h) compared to MIX-14 (99.17 ± 2.31 h) and BCWPRL (100.17 ± 2.52 h), there were no statistically significant differences between the three analyzed diets (MSE = 5.3, df = 6; IAEA-BY vs. BCWPRL *p* = 0.32; IAEA-BY vs. MIX-14 *p* = 0.57; and BCWPRL vs. MIX-14 *p* = 0.86).

### 3.2. Pupation Dynamics

A comparison of the total number of pupae collected at three collection events (24 h, 48 h, and 72 h after pupation onset—Table 1) showed a significant difference in the total mean number of pupae formed during the entire period of 72 h (MSE = 20.33, df = 6.0) between IAEA-BY and BCWPRL, as well as between BCWPRL and MIX-14 larval diets. The diet IAEA-BY resulted in the highest total mean number of produced pupae (3475.00 ± 179.67), but compared to MIX-14 (3399.67 ± 328.36), the differences were not significant (*p* = 0.94). A significantly lower number of pupae were obtained in the variant with a BCWPRL diet (2105.00 ± 78.10), compared to both the MIX-14 and IAEA-BY (*p* = 0.003 and *p* = 0.002, respectively). The quality of the IAEA-BY and MIX-14 diets was reflected in the similar number of formed pupae. Out of the initial population (4000 eggs), the majority of individuals that were fed with IAEA-BY and MIX-14 diets pupated (86.9% and 85%). In the same period, the percentage of pupated individuals fed with BCWPRL was much lower (52.6%). 

When the total mean number of pupae produced from 4000 eggs, collected 24 h after pupation onset, were compared (MSE = 40,263.0, df = 6.0), a significantly lower number of pupae were produced by BCWPRL (603.00 ± 187.85) than by the other two diets IAEA-BY (1748.33 ± 253.28) and MIX-14 (1371.67 ± 146.12), (BCWPRL vs. IAEA-BY *p*= 0.001, and BCWPRL vs. MIX-14 *p* = 0.008). The diets (IAEA-BY and MIX-14) did not differ significantly in the yield of pupae 24 h after pupation onset (*p* = 0.13).

The mean number of pupae 48 h after pupation onset (MSE = 29,934.0, df = 6.0) showed a significant difference between BCWPRL and MIX-14 (*p* = 0.03). The IAEA-BY diet did not reflect significantly on the number of pupae compared to both diets (BCWPRL vs. IAEA-BY *p* = 0.302, and IAEA BY vs. MIX-14 *p* = 0.247).

The pupae collected 72 h after pupation onset showed no difference between the three tested diets (MSE = 5327.2, df = 6.00, IAEA-BY vs. BCWPRL *p* = 0.993, and IAEA vs. MIX-14 *p* = 0.726, and BCWPRL vs. MIX-14 *p* = 0.788). 

### 3.3. Sieving Efficiency

The percentage of the pupae that passed through the sieve in relation to the total number of pupae collected after 24 h of post-pupation onset and processed by sieving did not significantly differ between the three larval diet variants applied: IAEA-BY, BCWPRL, and MIX-14 (MSE = 56,124, df = 6.0; Table 2). A comparison of the three pairs of diets (MIX-14 and IAEA; IAEA-BY and BCWPRL; and MIX-14 and BCWPRL) resulted in values of *p* higher than 0.05 (*p* = 0.863; *p* = 0.661; and *p* = 0.389, respectively), indicating that there were no significant differences in the portion of sieved pupae. 

Anyhow, when numbers of passed pupae were compared to the starting population (4000 eggs; Table 2, column Passed**), the percentage of successfully sieved pupae in the variant with BCWPRL diet (9.81 ± 3.35) was significantly lower than in IAEA-BY (23.77 ± 3.07, *p* = 0.006) but not significantly different compared to MIX-14 (16.92 ± 3.43, *p* = 0.074). Furthermore, the effects of the diets IAEA-BY and MIX-14 on the percentage of successfully sieved pupae were also not significantly different (*p* = 0.149).

When the total yields of pupae (passed and not-passed) out of the total number of eggs included in the experiment were considered, statistical analysis showed that both diets, IAEA BY (43.71 ± 6.33) and MIX-14 (34.29 ± 3.65), resulted in significantly higher percentage values of pupae than BCWPRL (15.08 ± 4.70), with the *p* values lower than 0.01 (*p* = 0.001 and *p* = 0.007, respectively). IAEA BY and MIX-14 were not significantly different in this parameter. 

### 3.4. Ratio of Males to Total Pupae Produced

Within the total number of pupae that passed through the sieve (sieving conducted 24 h after pupation onset), the majority were males. The diets provided the following percentages of males (mean ± Sd): IAEA-BY 93.85 ± 6.63%, BCWPRL 97.23 ± 3.61%, and MIX-14 96.36 ± 4.00%. 

Among pupae that did not pass through, sieve males were also dominant (IAEA-BY 83.24 ± 4.45%, BCWPRL 85.68 ± 7.23% and MIX-14 80.41 ± 17.73%). The next day (48 h), the portion of males decreased, but male pupae were still dominant in the BCWPRL (78.90 ± 9.70%) and MIX-14 (59.84 ± 8.44%) variants, while the portion of male pupae in IAEA-BY decreased below 50% (37.32 ± 23.50%). On the last day of pupal collection (72 h after pupation onset), female pupae dominated in all three tested diet variants, when the portions of males were as follows: 11.60 ± 3.56% in IAEA-BY, 40.52 ± 15.28% in BCWPRL, and 20.60 ± 9.77% in MIX-14.

For pupae formed 24 h after the pupation onset, the number of males (identified after the adults emerged) in relation to the total number of pupae was calculated for both categories of pupae—those that passed and those that did not pass through the sieve—while for pupae formed 48 h and 72 h after the pupation onset, only the total number of pupae collected (there was no sieving) were presented (Table 3). 

There were significant differences in the number of passed males that pupated within 24 h after the pupation onset between diets (MSE = 17,719.00, df = 6.00). TheThe BCWPRL diet diet provided a significantly lower (*p* = 0.01) number of males (356.33 ± 113.51) than the IAEA-BY diet (890.00 ± 110.01), but compared to the number of passed males in MIX-14 variant (629.67 ± 167.84) the difference was not significant (*p* = 0.10). Additionally, the number of males in the IAEA-BY variant was not significantly higher than MIX-14 (*p* = 0.12).

The number of males that did not pass through the sieve did not significantly differ no matter which diet was used (MSE = 62,456.00, df = 6.00, IAEA-BY vs. BCWPRL *p* = 0.12, IAEA-BY vs. MIX-14 *p* = 0.87, and BCWPRL vs. MIX-14 *p* = 0.22). 

When total number of male pupae collected at 24 h were compared (passed + not-passed), the BCWPRL diet showed a significantly lower number of males (533.00 ± 133.37) than the other two diet variants (MSE = 39,334.0, df = 6.00; IAEA-BY vs. BCWPRL *p* = 0.002; BCWPRL vs. MIX-14 *p* = 0.016). The IAEA-BY and MIX-14 diets were not significantly different (*p* = 0.142). 

For both of the next periods of pupae collection (48 h and 72 h after the pupation onset), the differences in the number of obtained male pupae were not significant between the three diets. 

On the second day of pupal collection, 48 h after pupation onset (MSE = 69,628.0, df = 6.00), the highest mean number of pupae was collected in MIX-14 trays (914.00 ± 266.04), then in BCWPRL (806.33 ± 135.43), and the lowest number was in IAEA-BY (480.00 ± 346.07), but the differences were not significant (IAEA-BY vs. BCWPRL *p* = 0.35, IAEA-BY vs. MIX-14 *p* = 0.19, and BCWPRL vs. MIX-14 *p* = 0.87). 

Similar comparison results were shown for the next day of pupal collection, 72 h after pupation onset. The differences in the number of male pupae between the tested diets were not significant (MSE = 5078.9, df = 6.00, IAEA-BY vs. BCWPRL *p* = 0.10, IAEA-BY vs. MIX-14 *p* = 0.61, and BCWPRL vs. MIX-14 *p* = 0.35). 

Finally, the total, cumulative numbers of male pupae produced in each diet variant during the three collection days were compared (MSE = 26,720.0, df = 6.00). It was demonstrated that the diet MIX-14 produced the highest mean number of males (2219.33 ± 221.39), but it was not significantly higher (*p* = 0.62) than the mean number of male pupae produced in the IAEA-BY diet (2090.67 ± 120.34). The BCWPRL diet produced a significantly lower total number of male pupae (1541 ± 129.09) than both the IAEA-BY (*p* = 0.01) and MIX-14 (*p* < 0.01) diets. 

Changes in protandry ratios dynamics (the average number of male pupae collected for a particular period/the average number of male pupae collected for all three periods) show that both IAEA-BY and MIX-14 supported the development of the highest number of males within 24 h after pupation, while this was postponed to the second day when larvae were fed with BCWPRL (as shown in Figure 2, Table 3). 

### 3.5. Flight Capacity of Males

There was no significant difference in flight capacities between the males reared from larvae fed with IAEA-BY (57.33 ± 16.04%), BCWPRL (61.33 ± 3.06%), and MIX-14 (56.67 ± 16.17%) (MSE = 61.86, df = 6.00). A comparison showed similar flight capacity values, with high *p* values obtained (IAEA-BY vs. BCWPRL *p* = 0.940, IAEA-BY vs. MIX-14 *p* = 0.998, and BCWPRL vs. MIX-14 *p* = 0.918). 

### 3.6. Longevity of Males

Regardless of which diet was used, the first males died 2 days after eclosion. The maximal longevity of 109 days was recorded in males that were fed as larvae with the IAEA-BY diet. The longest male lifespan recorded for the MIX-14 diet was 107 days, and it was 106 days for the BCWPRL diet. The IAEA-BY diet caused higher initial mortality among the males than the other two groups (Appendix A).

Additionally, the fastest decrease of survival after day 40 was recorded in a group of males that, in the larval stage, were fed with IAEA-BY (Figure 3). Kaplan–Meier survival curves showed a significant difference in terms of the survival rate between each of the tested groups (Figure 3, Table 4; Chi-square = 21.28, df = 2.00, and *p* < 0.01). Males that were in the larval stage that were fed a IAEA-BY diet had the shortest mean survival time. Males fed with the MIX-14 larval diet lived 8 days longer than those fed with the IAEA-BY diet and 2 days longer than males that were fed with BCWPRL larval diet. 

A comparison by Log rank test showed a significant difference in mean survival times between IAEA BY and BCWPRL (*ρ* = 0.02), and between BCWPRL and MIX-14 (*ρ* = 0.01). A difference in longevity was highly significant between IAEA BY and MIX-14 (*ρ* < 0.01). 

### 3.7. Cost of Alternative and Standard Diets

A calculation of the costs of 100 kg of the three larval diets was conducted based on the prices of each ingredient, reported by Bimbilé Somda et al. [39,46] and Khan et al. [45], and the amounts required according the recipes (see Appendix A). The IAEA-BY diet mixture (standard diet) is 2.28 times more expensive than BCWPRL and 5.30 times more expensive than MIX-14. The cost of BCWPRL is 2.32 times higher than MIX-14. Bovine liver powder is the costliest ingredient in both the IAEA-BY and BCWPRL diets, while in the MIX-14 vitamin mix it is the most expensive one.

## 4. Discussion

Optimizing *Ae. albopictus* mass production requires larval rearing methods that would provide high larval survival, fast and homogenous larval development, size homogeneity within the population, and synchronized pupation onset, which would, altogether, finally result in the production of high-quality adults. The evaluation of adult quality is based on some key parameters, such as longevity, flight ability, mating capacity, fecundity, and fertility [43,47,48,49]. When a larval diet provides the key parameters of high quality, the next critical question is whether the diet components are easy to procure and are not expensive. 

To test the quality parameters of the three larval diets under investigation, we evaluated the (a) time to earliest pupation (the time required from the submersion of eggs into water to the pupation onset); (b) the pupation dynamics; (c) the sex separation efficiency in the sieving procedure; (d) the ratio of males to total pupae produced 24 h, 48 h, and and 72 h after onset of pupation; (e) the flight capacity of the sieved males; (f) the longevity of the sieved males; and g) the cost of the diets. 

The duration of larval development can significantly affect the production costs. If the rearing requires a longer time, a lower number of production cycles will be feasible in a certain time, and the costs of labor, energy, and diet consumption per production cycle will increase. Using a larval diet that reduces the larval developmental time and favorizes early pupation onset could be an advantage, as long as the quality of the mosquitoes produced is not affected. Although IAEA-BY provided the shortest (97.17 ± 2.08 h) and BCWPRL (100.17 ± 2.52 h) the longest larval development, estimated through the time of earliest pupation, the difference of 3 h was not significant. All three diets provided faster development of larvae compared than the studies where larvae were fed with insect-based diets [39,41]. In addition, our results were similar to those obtained in the study where the larvae were fed with diets containing the same ingredients we used but in different proportions [38]. On the other hand, the study of Sasmita et al. [50], conducted on *Ae. Aegypti,* showed that the BCWPRL diet supported a shorter larval developmental time than the IAEA 2 diet (similar to IAEA-BY). 

The total number of pupae formed during the period of 72 h after pupation onset did not differ between IAEA-BY and MIX-14, but BCWPRL provided a significantly lower number of pupae than the other two diets. The total number of pupae (males and females) produced from 4000 eggs, collected on the first day (24 h after pupation onset), was significantly lower with BCWPRL than the other two diets, while the IAEA-BY and MIX-14 diets did not significantly differ. On the second day of pupal collection (48 h after pupation onset), the mean number of pupae recorded for the BCWPRL variant was still significantly lower than for MIX-14, while the IAEA-BY diet did not produce a significantly different number of pupae compared to the other two diets. On the third day (72 h after pupation onset), all the diets provided a similar number of pupae. In experiments performed by Sasmita et al., the BCWPRL diet also reflected a lower pupation rate than the IAEA 2 diet [50].

When males are collected once per production cycle and when controlling protandry and size dimorphism, one of the most important parameters for mass production is the number of pupae formed 24 h after pupation onset that pass through the sieve (the sexing procedure). The number of male pupae formed in the first 24 h represents the actual yield, i.e., the productivity of mass rearing for SIT. Only these pupae are then sterilized, and emerged sterile males are released. 

The number of pupae that successfully pass through the sieve is the crucial parameter in the sexing of pupae because the portion of obtained male pupae should be ideally >99% [51,52]. 

When the numbers of sieved pupae were compared to the initial population (4000 eggs), the portion of sieved pupae from the BCWPRL diet was significantly lower than the one from IAEA BY but not significantly different from MIX-14. The IAEA-BY and MIX-14 diets were not significantly different for this parameter. 

When total yields of pupae after 24 h for the IAEA BY, BCWPRL, and MIX-14 diets were compared, both IAEA-BY and MIX-14 demonstrated a significantly higher percentage of males than BCWPRL. 

Although the highest ratio of sieved pupae after 24 h to the total number of pupae collected at the same day provided by BCWPRL (65.02 ± 9.00%) could be considered advantageous for this diet, it is not so. The ratio of 24 h total pupae production to the starting egg population was significantly lower by BCWPRL than in both IAEA-BY and MIX-14 diet variants. Similar to our results, Sasmita et al. [50] observed a higher portion of *Ae. aegypti* male adults reared from BCWPRL fed larvae than from larvae fed by the IAEA 2 diet (comparable to IAEA-BY). 

In the sieving procedure conducted with pupae formed at 24 h after pupation onset, the majority of the pupae that passed through the sieve were males, as demonstrated for all three diet variants applied (IAEA-BY 93.85 ± 6.63%, BCWPRL 97.23 ± 3.61%, and MIX-14 96.36 ± 4.00%). Such results are in accordance with the results obtained by Puggioli et al. [38]. 

Considering the mean number of male pupae that passed through the sieve, a significantly lower value was obtained for the BCWPRL diet (356.33 ± 113.51) than the IAEA diet (890.00 ± 110.01). The difference in the mean number of males between the MIX-14 and IAEA-BY diet variants was not significant, as well as between MIX-14 and BCWPRL. 

Regarding the portion of male pupae recorded at the intervals of 48 h and 72 h, the BCWPRL diet resulted in a higher percentage of males than IAEA-BY, which indicates that the BCWPRL diet prolonged the development of males. Indeed, for the BCWPRL diet, the peak of male production was recorded during the second day of pupation (806.33 ± 135.43, 52.33%), which represents a disadvantage for mass production for SIT. Among pupae collected 48 h after the pupation onset, males still dominated in populations fed by BCWPRL and MIX-14 (78.90 ± 9.70% and 59.84 ± 8.44% of males, respectively) but not when larvae were fed by IAEA-BY (37.32 ± 23.50%). In samples collected 72 h after pupation onset, the number of males decreased in all diet variants, with the resulting portion of males in the sex ratio of 11.60 ± 3.56% in the IAEA-BY diet variant, 40.52 ± 15.28% in BCWPRL, and 20.60 ± 9.77% in MIX-14).

*Aedes albopictus* is generally considered a weak flyer. Higher mean distances travelled (MDT) for males were recorded in experiments carried out in Texas, USA (1100 m) [53] and in Switzerland (685 m) [54]. However, other studies conducted in Italy and La Reunion Island demonstrated lower MDT of *Ae. albopictus* males, which did not exceed the values of 154 m [55] and 67 m [56], respectively. Nevertheless, it is very important to determine the flight capacity of males produced in mass rearing procedures because their dispersal within the area of release and competitiveness with wild males in female seeking and mating depend on their quality condition (i.e., flight ability).

IAEA developed the flight test device to measure the quality of sterile male mosquitoes via their capacity to escape a series of flight tubes within two hours and demonstrated that survival and insemination rates could be predicted by the results of a flight test, with over 80% of the inertia predicted [32,33]. We used this novel tool as a quality-control method to evaluate the cumulative quality of males reared with different larval diets.

Our study showed that the applied larval diets did not reflect differently to the flight capacity of males. In any case, the best flight capacity was recorded for males of the BCWPRL variant (61.33 ± 3.06% of males successfully passed the test), followed by IAEA-BY (57.33 ± 16.04%) and MIX-14 (56.67 ± 16.17%), but there were no statistically significant differences. All three diets tested in our study demonstrated lower flight capacity than the results obtained for males reared with the insect-based larval diet (78.71 ± 3.51%) [41]. 

Males produced in a mass-rearing process aimed for SIT need to survive and mate as long as possible after their release. The longer lifetime of males may increase the chances to find more females to mate. All three types of the tested diets provided a relatively long mean survival time of males in laboratory conditions, ranging from 42.13 ± 20.29 (IAEA-BY) to 50.01 ± 22.64 (MIX-14) days. The mean survival time was significantly higher for MIX-14 males than for both the BCWPRL and IAEA-BY males. Males fed with the IAEA-BY larval diet had a significantly shorter mean survival time than males from the other two diet variants. Males fed with the MIX-14 diet lived 8 days longer than those that were fed as larvae with the IAEA-BY diet. 

Sasmita et al. [50] concluded that high energy reserves, provided by a BCWPRL larval diet, were reflected in higher adult male longevity compared to the IAEA 2 diet. Compared to the results of other authors [38] who tested the diets of a similar composition, all of the three diets tested in our study (IAEA-BY, BCWPRL, and MIX-14) provided a satisfying survival rate of males. However, the MIX-14 laval diet provided a significantly longer lifespan of produced adult males, which gives it the advantage over the other two diets. 

Besides the shortest mean survival time, males fed with the IAEA-BY larval diet had a higher initial mortality than the males produced on the other two diets. In the frame of SIT, high initial mortality might be a particularly detrimental indication for the survival and mating success of sterile males after release. Additionally, adult males, which in the larval stage had been fed with the IAEA-BY diet, showed the fastest decrease trend of survival after day 40. 

Although IAEA-BY is considered the standard diet and has successfully been used in the mass rearing process of *Ae. albopictus* for many years, this diet contains a high proportion of bovine liver powder (36%). Bovine liver powder is a very costly component (10 times more expensive than all the other ingredients needed to prepare IAEA-BY mixture) and is difficult to procure in many countries. The costs of the two alternative diets, which do not contain bovine liver powder, are more convenient than IAEA-BY (BCWPRL is 2.28 times, and MIX-14 5.30 times, less expensive). 

The bovine liver powder, as well as the much cheaper tuna meal, are both rich in proteins, vitamins, and fatty acids [36]. Brewer’s yeast, containing carbohydrates, is also used as a diet ingredient in the mass rearing of *Ceratitis capitata* (Wiedemann 1824) [57] and is required for the normal development of *Ae. aegypti* [58]. In the MIX-14 diet, bovine liver powder is replaced with chickpea. Previous studies demonstrated the high nutritional value of chickpea, which is rich in poly-unsaturated fatty acids, proteins, carbohydrates, B-group vitamins, and certain minerals [59,60,61,62,63,64]. Khan et al. [45] showed that chickpea alone provides the efficient larval development of *An. stephensi* Liston 1901. 

In the present study, the cheapest diet MIX-14 performed quite well and promoted itself as a possible candidate for the replacement of the good but expensive IAEA-BY larval diet, enabling cheaper sterile male production and the sustainability of SIT use in mosquito control. 

The diet MIX-14 is comprised of tuna meal, brewer’s yeast, vitamin mix, and chickpea. The first three ingredients are industrially produced, and their quality and composition can be kept at the same level. It is more difficult to provide a quality standard for chickpea, an agricultural product with a relatively high nutrient composition variation. Therefore, the reproducibility of the results obtained should be evaluated in the near future.

## 5. Conclusions

The integration of the sterile insect technique in the control of invasive mosquitoes might be an effective alternative way to increase vector control effectiveness. The high cost of mosquito mass production limits the area-wide application of the SIT worldwide, particularly in low- and mid-income countries, which are exposed to the highest risk of mosquito-borne diseases. The optimization of the larval production aims to decrease the cost of mass production. The different compositions of the three evaluated diets (IAEA-BY, BCWPRL, and MIX-14) did not have a significant influence on the duration of larval development, the dominance of males that pupated in the first 24 h, or males’ flight capacity. 

The BCWPRL diet provided the slowest development of males, a higher ratio of males than females that passed through sieve (but fewer pupae than the other two diets), a lower number of males that passed through sieve 24 h after pupation onset, and a lower number of males in total. 

The mean survival time was highest for MIX-14 males. Males that were fed (as larvae) with IAEA-BY had the shortest average survival time and higher initial mortality than the males produced on the other two diets. Two alternative diets cost less than IAEA-BY (BCWPRL is 2.28 times, and MIX-14 is 5.30 times, less expensive).

High-quality males produced from larvae fed by the cheapest diet (MIX-14) might support using it to reduce the rearing cost and provide cheaper sterile males for the sustainable use of SIT in mosquito control.

A prerequisite for including this diet in large-scale mosquito production is the testing of the reproducibility of the results obtained. MIX-14 comprises tuna meal, brewer’s yeast, vitamin mix, and chickpea. The first three ingredients are industrially produced, and their quality and composition could be kept at the same level. It is more difficult to provide a standard quality for chickpea, an agricultural product with a relatively high nutrient composition variation. 

## Figures and Tables

**Figure 1 insects-13-00504-f001:**
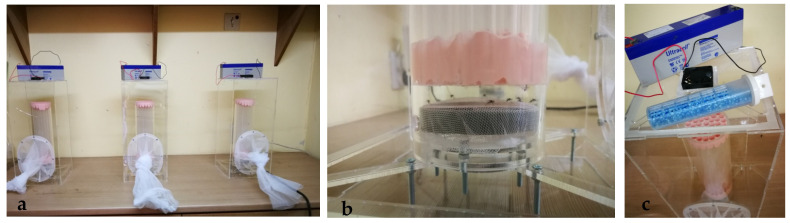
The flight capicity testing: (**a**) the simultaneous use of three IAEA flight devices. (**b**) The males introduced into the device cylinder. (**c**) The BG lure, energy source, and fan at the top of the device.

**Figure 2 insects-13-00504-f002:**
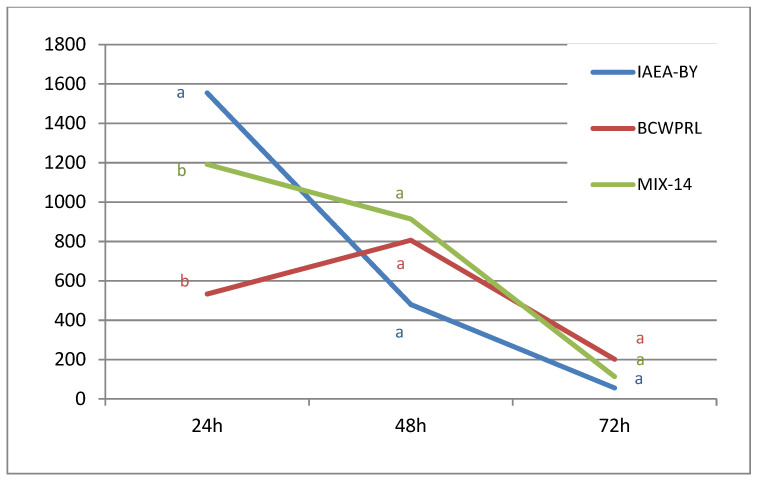
The mean number of male pupae collected 24 h, 48 h, and 72 h after the pupation onset (reared from 4000 eggs using three different larval diets). Values followed by the same letter are not significantly different (Tukey HSD test; α = 0.05).

**Figure 3 insects-13-00504-f003:**
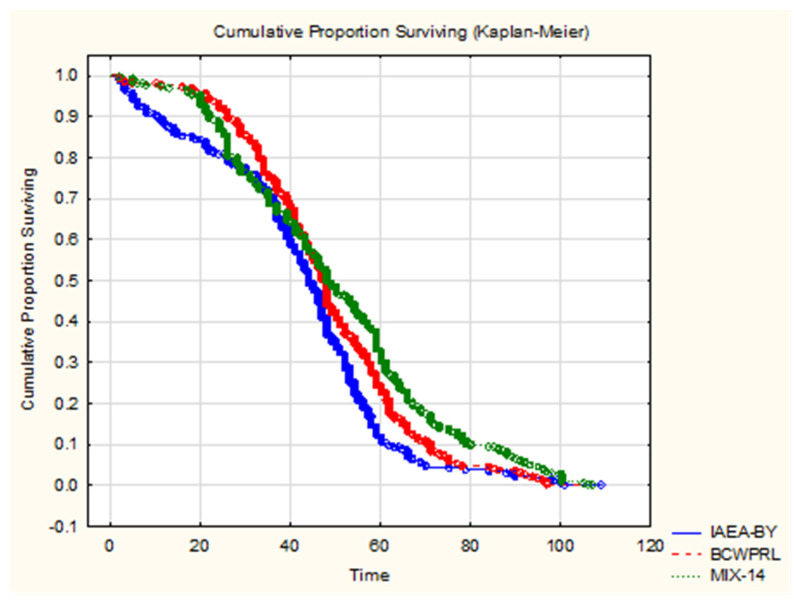
The survival curves of males fed in the larval stage with IAEA-BY, BCWPRL, and MIX-14 diets (all the presented data are complete; there are no censored data).

**Table 1 insects-13-00504-t001:** The number of pupae collected 24 h, 48 h, and 72 h after the onset of pupation, and the total number of pupae at the end of the experiment for three tested diets.

Replicates	24 h			48 h			72 h			Total		
	IAEA-BY	BCWPRL	MIX-14	IAEA-BY	BCWPRL	MIX-14	IAEA-BY	BCWPRL	MIX-14	IAEA-BY	BCWPRL	MIX-14
1	1524	674	1313	1452	983	1757	490	488	633	3466	2145	3703
2	2023	745	1538	1214	996	1457	422	414	450	3659	2155	3445
3	1698	390	1264	1089	1082	1305	513	543	482	3300	2015	3051
Mean Sd	1748.33 a 253.28	603.00 b 187.85	1371.67 a 146.12	1251.67 ab 184.41	1020.33 a 53.80	1506.33 b 230.00	475.00 a 47.32	481.67 a 64.73	521.67 a 97.74	3475.00 a 179.67	2105.00 b 78.10	3399.67 a 328.36

The values in the same row followed by the same letter are not significantly different (Tukey’s HSD test; α = 0.05). The statistical analysis was conducted separately for each of the collection intervals. The same applies for the total number of obtained pupae.

**Table 2 insects-13-00504-t002:** The ratio (%) of sieved pupae collected 24 h after pupation onset compared to the total number of pupae collected the same day (passed and not-passed through the sieve) *, and the ratio of sieved pupae number to the initial number of eggs submitted to hatching **, after feeding with three larval diets.

Diets	Passed *	Not Passed *	Passed **	Not Passed **	Total **
IAEA-BY	55.74 ± 14 a	44.26 ± 14 a	23.77 ± 3.07 a	19.94 ± 9.31 a	43.71 ± 6.33 a
BCWPRL	65.02 ± 9.0 a	34.98 ± 9.0 a	9.81 ± 3.35 b	5.25 ± 2.29 b	15.08 ± 4.70 b
MIX-14	50.24 ± 15 a	49.76 ± 15 a	16.92 ± 3.43 ab	17.36 ± 6.72 ab	34.29 ± 3.65 a

* Out of the total collected pupae at 24 h after pupation onset. ** Out of total number of potential individuals (4000 eggs). The values in the same column followed by the same letter are not significantly different (Tukey HSD test; α = 0.05).

**Table 3 insects-13-00504-t003:** The number of male pupae collected 24 h, 48 h, and 72 h after pupation onset and percentages in relation to the total male pupae obtained from variants with different larval diets.

Diet		Sieving Results (24 h)				Total Pupae per Collection Hours						Total
		Number		%		24 h		48 h		72 h		
		Passed	Not Passed	Passed	Not Passed	No.	%	No.	%	No.	%	
IAEA-BY	Mean Sd	890.00 a 110.01	664.67 a 320.80	57.25	42.75	1554.67 a 247.53	74.36	480.00 a 346.07	22.96	56.00 a 20.66	2.68	2090.67 a 120.34
BCWPRL	Mean Sd	356.33 b 113.51	176.67 a 64.42	66.85	33.15	533.00 b 133.37	34.59	806.33 a 135.43	52.33	201.67 a 96.46	13.09	1541.00 b 129.09
MIX-14	Mean Sd	629.67 ab 167.84	562.00 a 283.38	52.84	47.16	1191.67 a 197.37	53.69	914.00 a 266.04	41.18	113.67 a 74.19	5.12	2219.33 a 221.39

Values in the same column followed by the same letter are not significantly different (Tukey HSD test; α = 0.05).

**Table 4 insects-13-00504-t004:** The mean survival time of *Aedes albopictus* males that were fed different diets as larvae.

Diets	Mean ± Sd	Total Analyzed
IAEA-BY	42.13 ± 20.29 a	356
BCWPRL	48.19 ± 18.10 b	360
MIX-14	50.01 ± 22.64 c	358

Values followed by the same letter are not significantly different (Kaplan–Meier survival test, Chi-square = 21.28, df = 2.00, and *p* < 0.01). A statistical analysis included complete data; no data were censored.

## Data Availability

The data presented in this study are available in the article and in the Appendix A.

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
