# Peer review of "Optimization of Aedes albopictus (Diptera: Culicidae) Mass Rearing through Cost-Effective Larval Feeding"

_insects, 2022, doi:10.3390/insects13060504_

Round 1

Reviewer 1 Report

Dear authors, The research developed by you is important for the subject and must continue.
I made several comments and suggestions in the text to help structure it better.
However, my biggest concern is that only one experiment was performed with 3 replicates.
I believe this is insufficient to reach conclusions and experiments should be carried out with other generations of mosquitoes to ensure reproducibility.
I wish you success in the reformulation of this work and the next ones with this theme.
All the best

Author Response

Dear Reviewer 1,

On behalf of all the co-authors,

I would like to thank you for your extensive effort to review our manuscript. You gave us useful suggestions that helped us a lot to improve the manuscript. Also we would like to thank you for your opinion about the importance of the subject of our research. We tried to implement majority of your suggestions and we hope that you will find that the corrected version of our manuscript could satisfy the requirements of the Insects journal. In the lines below, you will find the answers to all of your comments.

Considering your biggest concern about the validity of our research results due to the experimental design (you suggested that different generations of mosquitoes should have been involved in the study), we would like to remind that our study was aimed to test the effects of the larval diets on different parameters important for mass rearing aimed to SIT. To achieve this objective, we actually believe that such type study should be done on the same generation of mosquitoes and under the same experimental conditions (as we did), to exclude the influence of any other unwanted factor. Our study was conducted on one mosquito generation, but in sufficient number of replications (3) and sufficiently high number of eggs initially involved in the study (4000 eggs per replicate, 3 diets, 3 replicates, 36000 eggs in total). The research was conducted at one experimental setup, and we think this was correct to justify the validity and of our results.

We also corrected the English language throughout the text of the manuscript and tried to make it clearer and more understandable.

In the lines below, you will find all our responses to your comments, one by one. We hope they will satisfy your expectations.

On behalf of the co-authors and myself personally,

Have our best regards,

Aleksandra Ignjatović Ćupina, corresponding author

_________________________________________________________________________________

Responses to Reviewer 1 comments and suggestions:

Lines 18-19:  Comment of Reviewer 1: BCWPRL resulted in the higher portion of sieved male pupae, but the  number of both pupae and adult males was lower . You said higher portion of male pupae and then mentioned the number was lower? Can you clarify? 

Authors`s Response: Authors agree that this sentence was not clear enough.

First part of sentence means that BCWPRL diet resulted with higher portion of males compare to females, and the second part is of the sentence is related to the comparison of the number of pupae between the three diet variants.

Therefore, the sentence is corrected to: „Application of BCWPRL resulted in the higher portion of sieved male pupae compared to females, but the obtained number of both pupae and adult males was lower and the development was slower compared to other two diets.”

Line 22: Comment of Reviewer 1: You did not mentioned before that they were alternative diet.

Authors`s Response: Authors deleted word “alternative”

Lines 77-82: Comment of Reviewer 1: Is it possible to cite the author here?

According the suggestion of the other Reviewer which we accepted, this part of the text was abbreviated and the sentence at lines 77-78 is deleted from the text.

Lines 108-109: Comment of Reviewer 1: Mosquitos were reared in Italy first and the eggs produced were sent to Serbia and reared again to establish a colony in Serbia? Please clarify and rewrite for better understanding.

Authors`s Response: Considering your question about mosquito colony: the eggs were produced in Italy in mass-rearing facility and sent to Serbia to conduct the experiment with diets, exclusively. Eggs were used only for experiment and not for the establishing the colony. Establishing of the colony wasn’t mentioned in the sentence.

The sentence is clarified with few changes.Produced eggs were shipped by express courier to the Laboratory of medical and veterinary entomology, University of Novi Sad, Serbia in order to perform the present study.”

Line 130:  Comment of Reviewer 1: So you performed the experiment only once with 3 replicates. Is it correct? The experiment was not repeated with different mosquito generations?  If not, I am not confident that you can reach accurety conclusions based in just one experiment.

Authors`s Response: The experiment was carried out on one generation of mosquitoes and in three replications for each applied larval diet variant. The present study was conducted with a sufficiently high number of eggs, sufficient number of replicates (enough for the statistical analysis) and it lasted several months. The study was conducted at one experimental setup, under same experimental conditions in order to exclude the influence of any other external unwanted factor. In relations to the objectives of our research, we actually believe that conducting the study on the same generation of mosquitoes and under same experimental conditions (as we did) was correct. Testing the effects of the diets on different generations of the same mosquito population was not the objective of our work. Therefore, authors consider the results are worth of attention for future similar studies.

Lines 135-136: Comment of Reviewer 1: What is this dilution you mention? You mixed broth and yeast in the 2ml of water and then tnrasfered to the jars? Please rewrite for better understanding.

Authors`s Response: The sentence is improved to make it more clear: “In order to stimulate the hatching, 0.25 g bacto nutrient broth (OXOID Ltd, UK) and 0.05 g yeast (Sigma-Aldrich Inc., MO) were dissolved in 2 ml of water and added into each jar.”

Lines 142-143: Comment of Reviewer 1: What was the rational for the diet concentrations added to the instars? Is it a convetional rearing protocol mentioned in other papers? Please clarify.

Authors`s Response: Thank you for the observation. Authors did not cite the procedure by mistake. The following sentence was added: “Diet amounts required per each tray and day were calculated based on the number of larvae, as described by Bellini et al. and Balestrino et al. [43, 44].”

Line 148: Comment of Reviewer 1: I suggest to put this information as supplementary data and not in the paper.

Authors`s Response: Authors agree with the comment. Table 1 is moved to Supplemental files (Supplemental Table 1) 

Lines 156-157: Comment of Reviewer 1: Insets refs to corroborate this statement.

Authors`s Response: Thank you for noticing that the citation is missing. Citation is added at the end of the sentence.

Lines 158-167: Comment of Reviewer 1: could you make a schematic drawing or photos that would help to better understand this whole pupae separation process for better understanding?

Authors`s Response: We agree on your comment. Authors corrected and improved the text of the entire paragraph to make in more understandable to the readers. Additionally, we prepared a supplemental figure (Supplemental Figure 1) showing the equipment and explaining the application steps of the sieving method.

Line 175: Comment of Reviewer 1: So the variable that you recorded here was total (or almost total) number of males pupae, right? You shoud mention this

Authors`s Response: We agree that the sentence is not enough precise. Pupae that passed through the sieve were expected to be males (usually majority of them are males, but not all of them, as you will see later in our results). For confirmation of the hypothesis we reared pupae till adult stage. Same procedure was conducted for those pupae which did not pass through the sieve (later in the text you fill find the sentence about this).

Therefore, at line 175, authors could not say that passed pupae were all males. However, we corrected the sentence at line 175 as follows: “The exact numbers of pupae which passed and not passed the sieve were recorded per each diet and replicate.”

Lines: 185-186: Comment of Reviewer 1: WHat is this for? You need a totral male/female ratio, right?

This part of the text (176-186) is rearranged in order to make it more understandable. The sentence at line 185-186 is now replaced with the following text: “The number of male and female specimens was recorded in order to calculate the portion of males in both “passed” and “not-passed” categories, as well as for calculation of the total number and portion of males developed from pupae collected 24h after pupation onset. Similarly, the portion of male pupae collected at 48h and 72h after pupation onset was calculated based on the number of emerged adults.”

Line 188: Comment of Reviewer 1: How many replicas for these tests? In fact, you made replicas of replicas and not new experiments. I'm not a statistician, but it could be a pseudo-replication

Authors`s Response: The number of replicates was always 3, and it also applies flight capacity and longevity test, as explained later in the following text related to the descriptions of the flight capacity and longevity test methods. Through the entire study we had 3 diets and 3 replicates for each of them (9 experimental units), and all mosquito parameters were studied in 3 repetitions.

We did not make pseudo-replicates, but from all pupae that “passed” through the sieve we made subsamples in each replicate in order to obtain desired number of adult males for: the flight capacity test (3 diets, 3 replicates, 50 males per each replicate) and longevity test (3 diets, 3 replicates, 120 males per each replicate,). Those pupae (subsamples) were put in separate cages, while the rest of pupae that “passed” the sieve were reared to adults in the third cage. Finally, all of them (all reared adults) were taken in consideration for sex ratio detection.

In the corrected version of the manuscript, the sentence at line 188 was replaced with the following detailed text:

“In order to obtain adult mosquitoes for further different testing purposes, pupae which successfully passed through the sieve (category“passed pupae”) were divided in sub-groups and placed separately in three cages. One cage with 70 pupae was aimed to obtain sufficient number of adult males for flight capacity testing, the second cage with 120 pupae was aimed for adult male longevity evaluation, while in the third cage the rest of “passed pupae” were placed and kept until the adult emergence. Adults which ecloded in all of the three cages were took in account for sex determination of the cumulative group “passed pupae”.

In the following part of the text, related to the descriptions of the flight capacity and longevity test methods the number of individuals and number of replicates is mentioned again.

Line 205: Comment of Reviewer 1: Is this longevity you mentioned before, right? Maybe longevity test…

Authors`s Response: Authors agree with the reviewer that the term “Longevity test” fits better than “Survival test”. The correction is made accordingly.

Line 214: Comment of Reviewer 1: My main concern here is that you have just one experiment with 3 replicates for each treatment… It seems that just one mosquito generation was tested.

Authors`s Response: The explanation is already given above- for comment indicated at Line 130.

Line 220: Comment of Reviewer 1: I suggest you change this passed and not passed pupae. Passed pupae are supposedly males and the not passed, females, right? Find a better designation for them.

Authors`s Response: It would not be correct to make those changes because as you will see in the results, among the pupae wich passed some were females and among the not passed a lot of them were males.

Line 229: Comment of Reviewer 1:  And how were analised? Chi-square?

Authors`s Response: We focused the statistical analysis on the number of male pupae produced daily (at 24h, passed and not passed, 48h and 72h) and their portion in the total number of male pupae produced within the three collection days, because in the SIT technique, the production of males is the target.

Anyhow, regarding the sex ratio, in the corrected version of the manuscript, in Results (3.4. Ratio of males to total pupae produced), we narratively presented % of males by mean values followed by standard deviation (basic statistics). Since no particular statistical method was used here, authors decided to delete the entire paragraph related to Statistical analysis.

Lines 237-239: Comment of Reviewer 1: If no differeces were find, then IAEA did not provides shortest pupation. Cite fig or table??

Authors`s Response: IAEA-BY diet provided the shortest development, but the difference compared to other two diets was not statistically significant. The sentence is corrected to make it clearer.

Since there were no significant differences detected, authors decided to present these results only in narrative. Authors agreed that a figure or a table is not necessarily needed.

Line 242: Comment of Reviewer 1: citation for table 2?

Authors`s Response: Table 2 is cited properly in the text at line 244. Authors believe that citing of tables in subtitles is not adequate. Anyhow, in the corrected version of the manuscript this table is now renamed as Table 1 (previous Table 1 is moved to Supplemental material, according the suggestions of the other reviewer)

Lines 245-246: Comment of Reviewer 1: Please, set the statistical info provided in the text. You mention significant differences, so provide P values and olso other test parameters. Please provide some data here instead of just saying that they were different.

Authors`s Response: Authors agree with the comment. The sentence is improved, results, test parameters, P values and mean values are added.

Lines: 248-252: Comment of Reviewer 1: You just said this above, isn’t it?

Authors`s Response: No, the lines 248-252 are related to Tab. 1 column 24h, while lines 245-246 are related to the column Total of the same table.

Line 266: Comment of Reviewer 1: Actually it not sex separation yet since you have data justo for passed and not passed pupae, right? Whn did you check if all passed were males or percentage of males?

Authors`s Response: Authors agree with the reviewers comment. The subtitle is renamed into “Sieving efficiency”

Lines 269-270: Comment of Reviewer 1: It is not usual to mention p values this way. Please rewrite.

Authors`s Response: Authors agree with the reviewers comment. Corrections are done accordingly.

Line 382: Comment of Reviewer 1: not cited in the text. (Tab. 3).

Authors`s Response: The reviewer is wrong, the table is cited in the first paragraph (under Sieving efficiency). In the corrected version, it is actually Table 2 now.

Lines 299-323: Comment of Reviewer 1: This is all about figure 2, right? the text is quite long and confusing> You need to summerize it and indicate the differences found in the figure and them just mention mais info in the text.

Authors`s Response: The text is elaborating the results presented in Table 4, as announced at line 298. (In the corrected manuscript, this is now Table 3). Authors corrected this part of the text to make it clearer and more understandable. Figure 2, demonstrating the tendencies of changes in the mean number of male pupae is given later. Therefore, in the corrected manuscript we put Table 3 before Figure 2)

Lines 335-336: Comment of Reviewer 1: Out of place

Authors`s Response: Author thanks the reviewer for observing this technical error. The pending sentence is actually the footnote of the table and somehow it remained at the inappropriate place. Now the sentence is deleted.

Line 341: Comment of Reviewer1: insert letters or number to indicate differences here. (Figure 2)

Authors`s Response: the letters indicating the differences are inserted in Figure 2

Line 343: Comment of Reviewer 1: Table 4 shows the same info of figure 2, right? Suggest you use just fig 2 and put the table in supplementary data.

Authors`s Response: We think that Table 4 is essential in presenting the related research results, numerically (means and  Sd) and in percentages, and also because it contains the results of statistical analysis (results of Tukey`s test). Therefore we would like to keep it in the manuscript. In the corrected version of our manuscript we improved this table according the suggestions of the other reviewer.

We would also like to keep Figure 2 because it visually demonstrate the protandry that is clearly expressed for the IAEA-BY and MIX-14 (the highest number of male pupae at 24h) but not for BCWPRL. Protandry is extremely important point for mass rearing in SIT.

Line 347: Comment of Reviewer 1: just sieved? Sounds strange written like this. Suggest you change it and explain in methods that just sieved males were used.

Authors`s Response: Authors agree with the reviewer, we changed the subtitle to “Flight capacity of males”. In materials and methods, we clearly explained that for the flight capacity test we used only males reared from sieved pupae (pupae which passed the mesh in the sieving procedure).

Lines 350-351: Comment of Reviewer 1:  remember what I said about the p values, ok? Where is the figure our table for this data? 

Authors`s Response: Authors agree with the reviewer regarding the first comment. The sentence is corrected, p values are given properly now.

Since there were no significant differences detected, authors decided to present these results only in narrative. Authors agreed that a figure or a table is not necessarily needed.

Lines 365-367: Comment of Reviewer 1:  I did not understand figure 3 and suggest that you just use figure 4 for to ilustrate mosquito longevity and also indicate the differences in the figure 4. Cite Fog 4 in the text.

Authors`s Response: Figure 3 represent the number of dead mosquitoes in intervals of 10 days (i.e. dynamics of adult male dying). Since row data is presented in this figure, we moved it to Supplemental material (entitled Supplemental Figure 2). In the corrected version of the manuscript, both Figures (Supplemental Figure 2 and Figure 3) are cited and commented properly .

Line 381: Comment of Reviewer 1: If you reaaly want to keep this and since you the alternative diets may be used in other countries in EU. I suggest you include the cost for other countries in EU. If not just mention the cost in the methods section.

Authors`s Response: The authors decided to move Table 6 to Supplemental materials (now: Supplemental Table 2), but prefer to keep the text about the comparison of the calculated costs of the diets in the manuscript. The cost of the diet is of a high importance for sustainable mass rearing. Our main idea was to test and compare the efficacy of the two cheap and one expensive diet and see if alternatives could work as efficiently as the standard diet, which is good but expensive and less affordable.

In the manuscript, we added the sentence “Calculation of the costs of 100 kg of the three larval diets was conducted based on the prices of each ingredient, reported by Bimbilé Somda et al. [39, 46] and Khan et al. [45], and the amounts required according the recipes (see Supplemental table 2)”. In the corrected manuscript, we also explained that for calculation of the costs we used the priced given in precious studies and cited the sources.

Providing data on the prices of the components in different countries would be too complicate, and probably the prices are quite variable, depending on the market situation.

Reviewer 2 Report

The invasive  and  medically and veterinary important mosquito-  Aedes albopictus  (and some other invasive species) has been rapidly  spreading over the world to areas where it was not  present till present. Even  in  the European conditions of mild climate  Ae. albopictus has been established in some areas and  northerly   it has reached almost 50°N latitude. Control of Ae. albopictus, owing to  its specific larval biotops requirements (human artefacts), is rather difficult. The control of Ae. albopictus is even  more (economically ) difficult in  tropical climates. The mosquito Ae. albopictus could be successfully suppressed by larviciding (e.g. by Bti) or other methods including the Sterile Insect Technique (SIT) in integrated mosquito management.

The SIT technique is based ion mass rearing, sterilization and release of males. Sterilized males then compete with males of the wild populations.

Mass rearing of the Ae. albopictus mosquitoes including use of cheap  larvae diet  and separation of  its males might be a  critical moment of the technique.

The authors tested  three types of mosquito larvae diet recipes in order to evaluate their economy and  flight capacity and longevity  from sieved pupae  hatched adults.

One of the tested diets  represented a candidate for replacing  an effective but expensive larval diet.

The submitted paper  deals with some important steps  ( mass rearing) of a  promising technique (SIT) of mosquito control. The paper is very well elaborated. I recommend it for publication in MDPI (Insects)

Author Response

Dear Reviewer 2,

On behalf of all the co-authors,

I would like to thank you very much for your effort to review our manuscript and for your positive opinion about the importance of our study and the quality of our manuscript in all of its parts.

On behalf of the co-authors and myself personally,

Have our best regards,

Aleksandra Ignjatović Ćupina, corresponding author

Reviewer 3 Report

See attached file.

Round 2

Reviewer 1 Report

Dear authors,

Thank you for considering the comments made by this reviewer and congratulations on the improvement in the quality of this second version of the manuscript.

I wish the group's research success.

Reviewer 3 Report

Well, I would still like to fight the authors over the question of "highly significant" (the fact that this error is so common that it has become a convention does not excuse it), but we can do that at a conference some time, I'm sure. In the mean time, the authors have addressed most of the concerns given in my original review. They have persuaded me on a couple of points and disagree with me on a few more, but I don't see the later as a barrier to publication.